# Why are song lyrics becoming simpler? a time series analysis of lyrical complexity in six decades of American popular music

**Michael E. W. Varnum**[1]*, **Jaimie Arona Krems**[2]*, **Colin Morris**[3]*, **Alexandra Wormley**[1], **Igor Grossmann**[4]*

**1** Department of Psychology, Arizona State University, Tempe, Arizona, United States of America,
**2** Department of Psychology, Oklahoma State University, Stillwater, OK, United States of America,
**3** Toronto, Canada, **4** Department of Psychology, University of Waterloo, Waterloo, ON, Canada

\* mvarnum@asu.edu (MEWV); jaimie.krems@okstate.edu (JAK); colin.morris2@gmail.com (CM); igrossma@uwaterloo.ca (IG)

**Data Availability Statement:** All data and reproducible code for analyses reported in the manuscript are available on the Open Science Framework (https://osf.io/qnsmj/).

## Abstract

Song lyrics are rich in meaning. In recent years, the lyrical content of popular songs has been used as an index of culture's shifting norms, affect, and values. One particular, newly uncovered, trend is that lyrics of popular songs have become increasingly simple over time. Why might this be? Here, we test the idea that increasing lyrical simplicity is accompanied by a widening array of novel song choices. We do so by using six decades (1958–2016) of popular music in the United States ($N$ = 14,661 songs), controlling for multiple well-studied ecological and cultural factors plausibly linked to shifts in lyrical simplicity (e.g., resource availability, pathogen prevalence, rising individualism). In years when more novel song choices were produced, the average lyrical simplicity of the songs entering U.S. billboard charts was greater. This cross-temporal relationship was robust when controlling for a range of cultural and ecological factors and employing multiverse analyses to control for potentially confounding influence of temporal autocorrelation. Finally, simpler songs entering the charts were more successful, reaching higher chart positions, especially in years when more novel songs were produced. The present results suggest that cultural transmission depends on the amount of novel choices in the information landscape.

## Introduction

Music is a human universal [1, 2], and it is known to influence cognition, affect, and behavior [3–5]. Because songs—and particularly popular song lyrics—can be so rich in meaning [6, 7], social scientists have long explored the ways that such lyrics intersect with some fundamental social processes, including identity formation and person perception [8–13].

More recently, social psychologists have begun to view music as a cultural product and to examine the ways that popular music lyrics reflect important aspects of psychology at the cultural level; the content in popular lyrics indexes changing norms, affect, and/or values [5, 14–19]. For example, DeWall and colleagues explored popular song lyrics as a "window into

**Funding:** The author(s) received no specific funding for this work.

**Competing interests:** The authors have declared no competing interests exist.

understanding U.S. cultural changes in psychological states" [5, pp. 200], finding that popular songs lyrics from 1980–2007 reflected an increase in self-focus and a decrease in other-focus.

Here, we demonstrate that popular music lyrics have become increasingly simple over time, and we test one possible explanation for this surprising trend, namely that the amount of novel song choices has increased.

## Novel song choices and lyrical simplicity

Several lines of evidence suggest that people may have baseline preferences for songs with simpler lyrics. One of the most widely known phenomena in psychology is the mere exposure effect, a phenomenon where repeated exposure to a non-aversive stimulus increases preference for it [20–22]. One implication of this principle for the present question is that simpler, more repetitive lyrics as these pieces essentially have this effect baked into them and thus may tend to be preferred all other things being equal. Further, songs with more repetitive lyrics may enjoy certain advantages in terms of information transmission as they are easier to remember [23] and likely easier to transmit with fidelity [24–26]. Further, recent work has shown that naïve listeners find simpler, more repetitive pieces of music to be more enjoyable, engaging, and memorable [27, 23].

Why might pop songs become lyrically simpler in times when more new songs are produced? Theory and research from diverse literatures suggest that songs with simpler lyrics might be especially successful when there are more new songs to choose from. First, humans are cognitive misers. People have limited information-processing capacities [28], and are known to conserve mental resources [29]. Consequently, humans often use shortcuts in decision-making [30, 31]. For example, when confronted with the task of evaluating persuasive messages and/or complex decision environments, people are more likely to use heuristics, peripheral cues, and other automatic cognitive processes to evaluate these messages if cognitive resources are limited in some fashion [32, 33]. Thus, when there are more products to be evaluated, people may increasingly prefer simpler products as they may require less mental effort to engage with. The mere exposure effect might also have a greater influence on decision making in such contexts as well, given that it too can be thought of as a heuristic or even instinctive evaluation. Further, across real-world studies and in-laboratory experiments, when people are confronted with a greater number of options to choose from, they are more likely to choose simpler, less cognitively demanding products [34]. Taken together, this work suggests that pop songs on average might become lyrical simpler in times when people are exposed to greater amounts of new songs and that success of such songs might be more strongly linked to lyrical simplicity in such times.

Here, we test the hypothesis that the trend toward increasingly simple popular music lyrics might be accompanied by the increasing number of songs released each year, using six decades' worth of song data. We also do so while including a number cultural and ecological control variables, as prior work demonstrates that well-studied ecological features, such as resource levels, pathogen threat, and sources of external threat (e.g., climatic stress, armed conflict) can impact markers of cognition and behavior at the cultural-level [35–38], and might plausibly affect preferences for simplicity in aesthetic products. For example, both resource scarcity and pathogen prevalence have been associated with conformity, innovation, and creativity in prior work [35, 39, 40].

## Methods

We gathered cross-temporal data covering a period of six decades (1958–2016) on lyrical compressibility (as an index of simplicity/complexity of song lyrics), amount of novel songs

produced (as an index of available novel song choices), and ecological, socioecological, and cultural variables linked to patterns of cultural change in previous research or plausibly related to trends in aesthetic content.

## Lyrical compressibility of successful music

We gathered data from 14,661 songs that entered the Billboard Hot 100 charts spanning the period from 1958 (the charts inception) to 2016. The Billboard Hot 100 tracks the 100 most popular songs each week based on music sales, radio airplay, and internet streaming. To operationalize lyrical complexity (vs. simplicity), we estimated text compressibility. By operationalizing complexity via a compressibility index, we avoided some of the conceptual ambiguity associated with operationalization of complexity in prior research [40–42]: Whereas multi-purpose use of a single product may reflect product's complexity from the operational standpoint, it may also represent greater simplicity from the standpoint of consumer psychology. Further, song lyrics are tractable to work with when using an automated compression algorithm.

Compressibility indexes the degree to which song's lyrics have more repetitive and less information dense, and thus simpler, content. We used a variant of the established LZ77 compression algorithm. In brief, the LZ77 algorithm works by finding repeated substrings and replacing them with 'match' objects pointing back to the string's previous occurrence. A match is encoded as a tuple $(D, L)$, with $D$ being the distance to the substring's previous occurrence, and $L$ being its length. We treated these matches as costing 3 bytes. This way, a repeated string only leads to space savings if it is of at least length 4, and longer repetitions lead to greater relative savings. Given a song $S$, and the set of matches $M$ produced by the LZ77 algorithm when applied to that song, its compressed size is therefore:

$$compsize(S) = |S| - \sum_{(D,L) \in M} L - 3$$

Where $|S|$ is the original size of the song's lyrics, measured in characters/bytes. The compression ratios of songs in our dataset (i.e., $|S|/compsize(S)$) followed an approximately log-normal distribution, so we operationalized compressibility as the logarithm of this ratio:

$$compressibility(S) = log(\frac{|S|}{compsize(S)})$$

We used the LZ77 compression algorithm because of its intimate connection to textual repetition. Most of the byte savings when compressing song lyrics arise from large, multi-line sections (most importantly the chorus, and chorus-like hooks). Another significant contributor are multi-word phrases, which may be repeated in variations across different lines for poetic effect (e.g. the anaphoric verses in Lady Gaga's *Bad Romance*: "I want your ugly / I want your disease / I want your everything . . ."). The compression may make use of repeated individual words, or even sub-word units that repeat (perhaps incidentally), but their contribution to the overall compressibility is low.

Higher compression scores signify more repetition and therefore higher simplicity. A score of 0 means no compression was possible (e.g. if the input were random noise), a score of 1 means a 50% reduction in size, a score of 2 means a 75% reduction, and so on. For example, Daft Punk's 1997 song "Around the World" repeats the title 144 times and has a compressibility score of 5.42 (the maximum in this sample). Nat King Cole's "The Christmas Song" (1961) has a low compression score of 0.11.

We computed mean compressibility for each year based on all songs that entered the Hot 100 charts in a given year for which we were able to scrape lyrics (1958–2016). Because we

used an automated procedure for song scraping, which depends on the readability of the song lyrics, the percentage of songs scraped varied between 27% of top 100 songs in 1958 and 91% of songs in 2015 ($M$ = 57%, $Md$ = 57%, $SD$ = 19%). Because percentage of scraped songs has been increasing over time, and correlated with the compressibility index, $\tau$ = .73, $p < .001$, in additional analyses we controlled for this trend.

## Song success

Some of the theoretical positions we draw on to evaluate possible reasons for changes in lyrical complexity suggest that more compressible songs may be more likely to be successful. To evaluate this proposition, we additionally gathered data on the highest position of each song in the sample achieved on the Billboard charts.

## Novel music production

In the spirit of the multiverse analyses [43], we used three separate indicators to assess the amount of new music to which people are likely exposed in a given year. For each year (1958–2016) we computed the total number of songs which made the Hot100 chart, the number of musical releases per year according to Discogs (Discogs.com), and the number of Wikipedia entries about songs first published or performed each year (Wikipedia.org).

## Possible ecological drivers of cultural change in aesthetic preferences and music production

We assessed a range of well-studied socioecological factors (e.g., resource levels, pathogen threat, sources of external threat), which could plausibly bear on aesthetic preferences or might affect lyrical simplicity (and whether the predicted association between novel music production and simplicity holds even controlling for these or other ecological and cultural variables discussed below). *Resource scarcity* has been linked to greater conformity [39] and cross-temporal work has found that greater resource levels are linked to more innovation and creative output [40] and less conformity [44, 45]. Higher levels of *infectious disease* have also been linked to more conformity [46, 47], traditionalism [48], and tight social norms [35, 49]. *External threats*, due to climate or war, have also been linked to more traditional outlooks and tight social norms [49], which might similarly bear on trends in lyrical simplicity. We thus included publicly accessible data indexing these factors GDP per capita, GDP growth, unemployment, pathogen prevalence, climatic stress, and participation of the US in major armed conflicts. The data used in our analyses covered the years 1958–2016. Data on GDP per capita and GDP growth were gathered from macrotrends.net, and data on the other markers came from Varnum & Grossmann [50] and updates from the original data sources used in that publication.

We also explore the possible impact of other socioecological factors that might plausibly affect lyrical simplicity. One might speculate that *immigration* could drive increases in lyrical simplicity. For example, simpler lyrics in American pop songs might be linked to shifts in the amount of people for whom English may not be a first language. In a similar way, it might be that *ethnic fractionalization*, so far linked to changes in individualism and uniqueness over time [51], may also increase preferences for, memory of, and/or dispersal of simpler, more repetitive lyrics, as such content would be easier to convey and understand to a wide range of audiences. To assess the possibility that a rise in simpler English lyrics might be linked to shifts in the amount of people for whom English may not be a first language, we used data on the number of green cards issued from the Department of Homeland Security as a marker of immigration. To assess possibilities linked to ethnic fractionalization, we used data on ethnic fractionalization from the US Census Bureau.

Research on the consequences of *residential mobility* also suggests that perhaps this variable might also affect lyrical trends. Previous studies have linked residential mobility to greater susceptibility to the mere exposure effect and greater preference for familiar cultural products [52]; thus, it may be that mobility is also linked to temporal variations in lyrical complexity of pop songs. To assess residential mobility, we gathered data on percentage of the US population that changed residence within the US from the US Census Bureau.

At the same time, a simpler variable might also be driving this effect. Perhaps products that succeed with a larger audience are merely simpler, akin to a lowest common denominator effect. Because the U.S. population grew substantially in recent decades, we also test whether population trends might be associated with lyrical simplicity. Thus, we also gathered data on the total size of the US population from macrotrends.net to explore population size.

## Cultural factors

Prior work has found conservatives show a preference for simple and unambiguous art, speech patterns, and literature [53–57] (though see also Conway et al., 2016 [58]). Thus, one might suspect that possible changes in conservatism could be driving lyrical simplicity. Somewhat similarly, other evidence suggests that cross-cultural differences in aesthetic preferences and expression are linked to orientations toward collectivism [59, 60]. Thus, we also gathered data on indicators of *conservative ideology*, operationalized conservatism as the average percent of annual survey respondents in Gallup polls identifying as conservative, and we included as an index of cultural level collectivism based on frequency of collectivism related words in the Google Ngrams American English corpus [45].

## Analytic procedure

Where possible, we use non-parametric ordinal-level measures of correlation or partial correlation (*Kendall*'s rank correlation coefficient τ), which provides estimate of similarity of the orderings of the data when ranked by each of the quantities. Since Fechner's initial work on time series analyses, *Kendall*'s τ has been a preferred metric for examining cross-temporal relationships [61]. It provides a conservative estimate, which is preferred because time series data is rarely normally distributed. Results were comparable when we used Pearson's *r* or partial Pearson correlations. In the initial step, we examined zero-order relationships between each of the three indices of available novel song choices and average lyrical compressibility of popular songs. Next, we created a composite index of novel song choices and assessed the robustness of the hypothesized link between amount of novel song choices and average lyrical compressibility of popular songs by controlling for a host of ecological, socioecological, and cultural factors that might plausibly influence cultural level success for simplicity vs. complexity. Our chief analyses focused on a set of corrective analyses, in which we controlled for the possibly spurious nature of the relationship between our key time series due to temporal autocorrelation.

Given the range of possibilities of correcting for temporal autocorrelation, we opted to perform three different types of analyses that correct or account for the possibility that observed relationships might be spurious as a function of autocorrelation in the time series. First, we computed adjusted significance thresholds based on the Tiokhin-Hruschka procedure [62]. Second, we detrended our novel song production and lyrical compressibility time series by residualizing for year and assessed the correlation between our detrended variables. Finally, for central univariate and multivariate analyses, we used an automated auto-regressive integrated moving average forecasting model (auto.ARIMA) to assess the relationship between novel song choices and lyrical compressibility [63]. This technique involves a machine learning algorithm that tests a number of different possible models which vary in autoregressive

components, differencing, and moving average components, as well as whether they include an exogenous predictor. Additionally, we used auto.ARIMA to generate a forecast for future patterns of lyrical compressibility (2017–2046).

For multivariate analyses we entered multiple predictors of lyrical compressibility over time. To avoid multicollinearity and overfitting (and due to limited number of units at the yearly level of analysis), we first aggregated covariance scores attributed to additional socioeco-logical and cultural factors (see Table 1) by performing a principal component analysis on these covariates and saving component scores for further multivariate time series analyses. The first principal component explained 50% of the variance in the covariates, with strong loadings (absolute value >.85) for Population Size, GDP/capita, Residential Mobility, Patho-gen Prevalence, Ethnic Heterogeneity and Immigration, moderate loadings for Armed Con-flicts (.49) and weak loading of GDP growth (.44). Other covariates (Climatic Stress, Unemployment, Conservatism, Collectivism) showed very weak loadings (.21 < absolute value ≤ .27). Next, we entered both yearly music production scores and covariate-PCA scores as independent predictors of lyrical compressibility, simultaneously accounting for the time series structure in the data.

## Data availability

All data and reproducible code for analyses reported in the manuscript are available on the Open Science Framework (https://osf.io/qnsmj/).

**Table 1. Correlations with average lyrical compressibility.**

| | Variable | *Kendall's τ* | *Kendall's τ* (Detrended) |
|---|---|---|---|
| *INFORMATION LANDSCAPE* | Music Production | .714*** | .222** |
| *ECOLOGICAL* | GDP per capita | .733*** | .044 |
| | GDP growth | -.260** | -.073 |
| | Unemployment | .051 | .135 |
| | Pathogen Prevalence | -.490*** | .324** |
| | Climatic Stress | -.118 | .050 |
| | Armed Conflict | .229* | .063 |
| *SOCIO-ECOLOGICAL* | Immigration | .563*** | -.155 |
| | Ethnic Heterogeneity | .737*** | -.066 |
| | Residential Mobility | -.692*** | -.230* |
| | Population Size | .726*** | .135 |
| *CULTURAL* | Conservatism | -.019 | -.287* |
| | Collectivism | -.225* | -.124 |

*p < .05,

** p ≤ .01,

*** p ≤ .001.

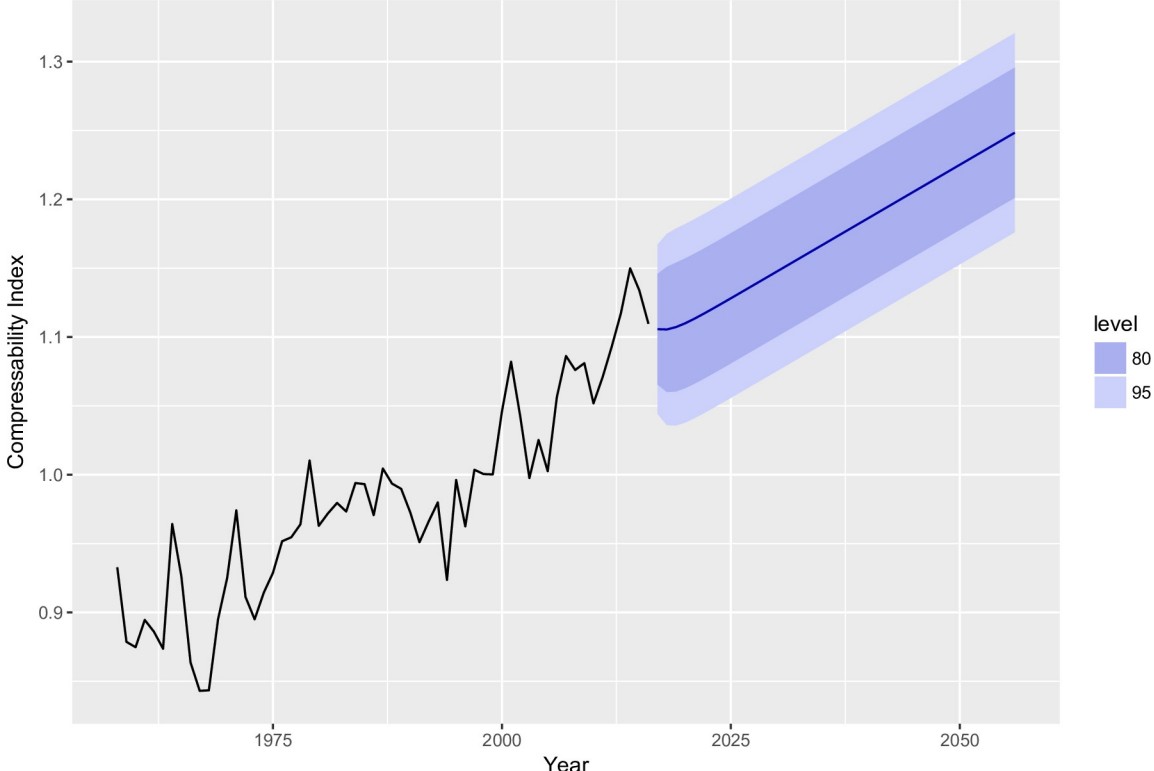

**Fig 1. Change in lyrical compressibility, along with a music production-based forecast for future lyrical compressibility from regression with ARMIA (1,0,0) and index of novel song choices as an exogenous predictor.** Light purple indicates 95% confidence bands, dark purple indicates 80% confidence bands.

## Results

### Indicators of novel song choices and average lyrical compressibility

As Fig 1 indicates, mean lyrical compressibility (i.e., simplicity) of songs increased over time, *Kendall's* τ = .726, *p* < .001, as did number of songs making the Hot 100 charts per year, *Kendall's* τ = .425, *p* < .001, number of music releases according to Discogs per year, *Kendall's* τ = .973, *p* < .001, and number of Wikipedia entries for songs by year of publication, *Kendall's* τ = .871, *p* < .001.

### Analyses of the composite index of novel song choices

Hot100 songs, Discog music releases, and Wikipedia song entries were highly correlated, .41 < *Kendall's* τ's ≤ .87, and formed a single principle component with highest loadings by the Wikipedia song entries (.98), and weakest loading by the Hot 100 songs (.88). To avoid multicollinearity, we used component scores for further analyses. Overall, this index of novel music production was strongly positively related to compressibility, *Kendall's* τ = .714, *p* < .001. Consistent with our predictions, mean lyrical compressibility per year was positively correlated with amount of novel music produced per year as operationalized by three distinct indicators, *Kendall's* τ (*n* songs in *Hot 100* charts/year) = .429, *p* < .001, *Kendall's* τ (*n Discogs* music releases / year) = .721, *p* < .001, *Kendall's* τ (*n* Wikipedia entries about songs/year) = .680, *p* < .001.

## Relationships between socioecological factors and compressibility

Although several ecological dimensions were associated with changes in average lyrical compressibility over time (see Table 1), these relationships were often in the opposite direction that prior research or theorizing would suggest. For example, there were significant negative correlations between GDP per capita and pathogen prevalence and average lyrical compressibility. Further, our two cultural variables were either unrelated to lyrical compressibility (conservatism) or correlated in the opposite of the predicted direction (collectivism). We did observe theoretically sensible relationships between compressibility and residential mobility, immigration, ethnic fractionalization, and population size. However, when controlling for the potentially confounding effect of temporal auto-correlation by residualizing out the effect of year, only three of these relationships are statistically significant, and only the relationship between pathogen prevalence and average lyrical complexity remains in a theoretically sensible direction (see Table 1).

## Robustness analyses: Control variables

This PCA-based composite index of music production remained significantly related to lyrical compressibility when including percentage of scraped songs/year as a covariate, *Kendall*'s $\tau_p$ = .261, *p* = .003. Further, it remained significant when controlling separately for each of the 12 specified control variables, .220 < *partial Kendall's* $\tau$'s < .770, p's < .02 (see Table 2 for details). Full correlations between these variables are presented in S1 Fig.

## Robustness analyses: Auto-correlation

Importantly, the correlation between this composite index of novel song choices and average lyrical compressibility remained significant when adjusting significance thresholds using the Tiokhin-Hruschka method to account for observed auto-correlation in the two time series, *r* =

**Table 2. Partial correlations between novel music production index and average lyrical compressibility.**

|  | Control Variable | *Partial Kendall's* τ Novel Music Production & Lyrical Compressibility |
|---|---|---|
| ECOLOGICAL | GDP per capita | .248** |
|  | GDP growth | .695*** |
|  | Unemployment | .753*** |
|  | Pathogen Prevalence | .596*** |
|  | Climatic Stress | .710*** |
|  | Armed Conflict | .696*** |
| SOCIO-ECOLOGICAL | Immigration | .539*** |
|  | Ethnic Heterogeneity | .267** |
|  | Residential Mobility | .436*** |
|  | Population Size | .231** |
| CULTURAL | Conservatism | .670*** |
|  | Collectivism | .610*** |

*p < .05,

** p ≤ .01,

*** p ≤ .001.

.877, $_{corrected}p < .001$. As an alternative method for dealing with autocorrelation, we also detrended the time series by residualizing out the linear impact of year. The correlation for our detrended variables remained significant, *Kendall's* $\tau = .222$, $p = .010$.

Given the time series nature of our data, another way to test the hypothesized link between amount of new songs available and average compressibility of these songs while also addressing the issue of autocorrelation can involve an automated ARIMA algorithm (auto.ARIMA) within the forecast package [64] in *R* 4.0.0 [65]. This machine-learning algorithm inspects the time-series data to fit the optimal forecasting function. The auto-regressive (*AR(p)*) component refers to the use of past values in the regression equation for the series Y. The auto-regressive parameter p specifies the number of lags used in the model. A moving average (*MA(q)*) component represents the error of the model as a combination of previous error terms $e_t$. The order *q* determines the number of terms to include in the model. ARIMA models are well-suited for long-term time series, such as the historic patterns in the present data. The automated algorithm within the forecast package searches through combinations of order parameters and picks the set that optimizes model fit criteria, comparing Akaike information criteria (AIC) or Bayesian information criteria (BIC) of respective models. Notably, the automated forecasting approach allows us to specify an exogenous predictor such as novel song choices, such that the automated function can evaluate the extent to which this exogenous predictor improves the fit above and beyond the decomposition of the time-series of the dependent variable. In other words, the automated function provides a conservative way to see whether an exogenous predictor such as the novel song choices index improves accuracy in forecasts of the lyrical compressibility. If the final model selected by auto.ARIMA includes our putative exogenous variable (in this case amount of novel song choices), then this suggests that this variable helps the model to achieve optimal fit to the data.

The results of this automated forecasting procedure indicated that a model with a positive autoregressive component, $B = .527$, $SE = .124$, and a positive contribution of the novel music production index, $B = .059$, $SE = .008$, provides the best fit to the data:

$$y_t(\text{lyrical compressibility function}) = .983 + .527y_{t-1} + .059x + e_t$$

This model estimation suggests that the index of novel song choices contributes to average lyrical compressibility above and beyond the temporal autocorrelation observed for average lyrical compressibility. Further, the coefficient for the index of novel song choices was statistically significant, $z = 6.95$, $p < .001$.

We also ran an alternative set of auto.ARIMA analyses where we set novel song choices as the dependent variable and average lyrical compressibility as an exogenous predictor. The results of this automated forecasting procedure indicated that a model with two positive moving average components, $B = 1.176$, $SE = .242$, and $B = .487$, $SE = .164$, and a positive contribution of average lyrical compressibility, $B = 5.067$, $SE = 2.207$, provides the best fit to the data:

$$y_t(\text{novel music production function}) = -4.991 + 1.176\varepsilon_{t-1} + 0.487\varepsilon_{t-2} + 5.067x + e_t$$

The coefficient for lyrical compressibility was statistically significant, $z = 2.30$, $p = .02$.

Comparison of the Akaike Information Criterion (AIC) and Bayesian Information Criterion (BIC) values for our primary and alternative models suggest that our primary model with novel song choices as an exogenous predictor and lyrical compressibility as the dependent variable, AIC = -235.84, BIC = -227.53, is superior to the alternate model with lyrical compressibility as an exogenous predictor and novel song choices as the dependent variable, AIC = 58.36, BIC = 68.75.

## Robustness analyses: Controlling for percentage of scraped songs

Because of a positive association between lyrical compressibility and percentage of scraped songs per year, we performed a separate set of analyses in which we first regressed out the effect of sampling (% of scraped songs/year) on lyrical compressibility and performed an auto. ARIMA analysis on the residuals. Results of a model on the residuals with music production as a predictor indicated a significant effect of music production, $B = .799$, $SE = 0.046$, $z = 17.32$, $p < .001$, suggesting that the effect songs even when accounting for the possible change in sampling.

## Multivariate analyses

In another set of control analyses, we performed an auto.ARIMA analysis, in which we included the PCA factor formed by all socio-ecological covariates as a second covariates. By comparing the magnitude of the effect from this first principal component (which was chiefly driven by ecological variables) and music production index, we can assess the relative contribution of the music production index via-a-vis other socio-ecological covariates. The results of this automated forecasting procedure indicated that a model with a positive autoregressive component, $B = .513$, $SE = .118$, a significant positive contribution of the novel music production index, $B = .038$, $SE = .016$, $z = 2.37$, $p = .018$, and a non-significant positive trend formed by ecological covariates (and chiefly reflecting economic and population growth), $B = .026$, $SE = .016$, $z = 1.61$, $p = .108$, provides the best fit to the data:

$$y_t = .981 + .513y_{t-1} + .038(\text{music production}) + .026(\text{ecological covariates}) + e_t$$

This model estimation suggests that the index of novel song choices contributes to average lyrical compressibility above and beyond the temporal autocorrelation as well as other ecological covariates observed for average lyrical compressibility. Moreover, the effect of music production on lyrical compressibility was stronger than other feasible covariates explored in the present dataset.

## Exploratory song-level analyses

In exploratory analyses we evaluated how lyrical compressibility is associated with song success, and whether this relationship was stronger in time periods when more novel music was produced. Given that we shifted focus to song-specific data, we utilized a multi-level framework via *lme4* package in *R*, with songs' chart position and lyrical compressibility scores nested within years. Preliminary auto.ARIMA analyses on the yearly aggregate data indicated that a model with no auto-regressive components but a linear trend would show the best model fit. Therefore, in the first multi-level model we included year as a proxy for a linear trend as well as compressibility X year interaction as predictors of song success. Both year and lyrical compressibility were mean-centered prior to analyses. This multi-level model showed a good overall model fit, $R^2 = .05$, with 3.9% of the variance explained by fixed effects. Results indicated a significant effect of year, $B = 0.318$, $SE = 0.031$, $t(df = 57.29) = 10.23$, $p < 001$, suggesting that over time songs included in the sample on average had a lower chart rank—a typical regression to the mean effect. Importantly, more compressible songs showed significantly higher rank in the charts, $B = -9.321$, $SE = 0.661$, $t(df = 14640.88) = 14.10$, $p < .001$, and this effect was particularly pronounced for more recent years, compressibility X Year interaction, $B = -0.105$, $SE = 0.039$, $t(df = 14581.41) = 2.71$, $p = .007$.

In the second step, we added mean-centered yearly music production index as a second covariate, along with a music production X compressibility interaction. Based on prior auto.

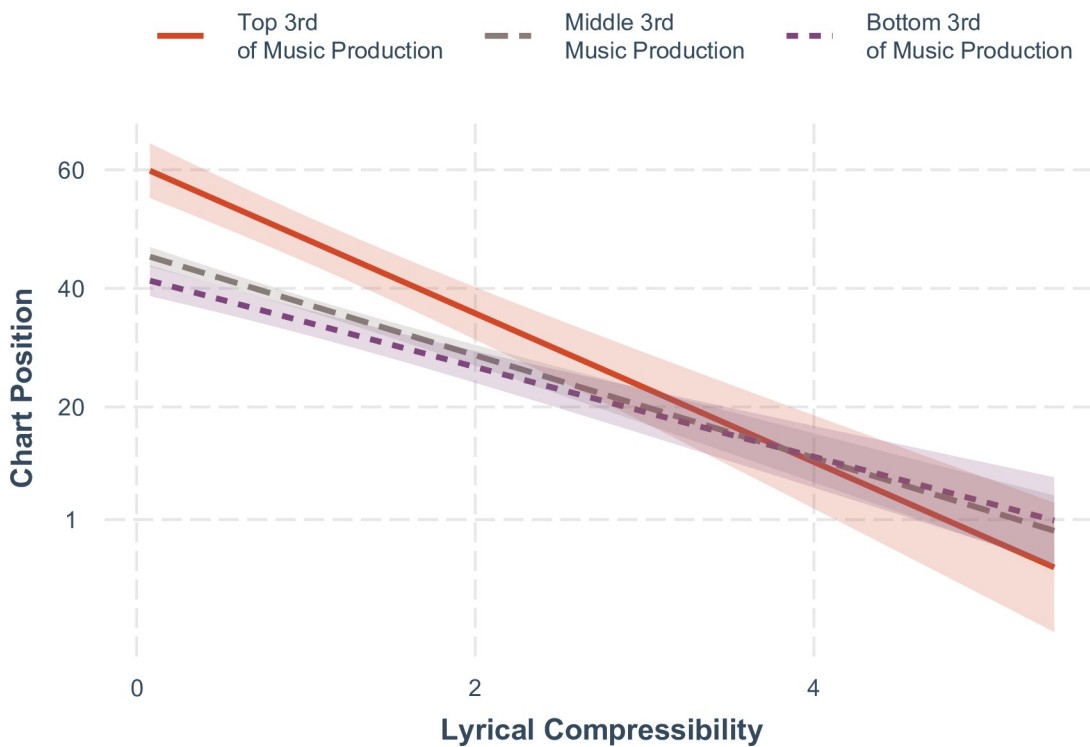

**Fig 2. Relationship between lyrical compressibility and chart position for years differing in music production volume.**
Confidence bands indicate 95% around the estimate.

ARIMA results, we also included linear effect of year to account for the trend in the chart position. This multi-level model also showed a good overall model fit, $R^2$ = .06, with 4.7% of the variance explained by fixed effects. More compressible songs showed significantly higher rank in the charts, $B$ = - 9.353, $SE$ = 0.657, $t(df$ = 14819.95) = 14.23, $p <$ .001. Also, average chart position of songs was higher in years with a greater volume of songs produced, $B$ = 6.141, $SE$ = 1.280, $t(df$ = 53.76) = 4.80, $p <$ .001. Moreover, as Fig 2 indicates, lyrical compressibility was more strongly associated with song success in years with greater volume of produced songs, compressibility X music production interaction, $B$ = - 2.170, $SE$ = 0.648, $t$ $(df$ = 14781.15) = 3.35, $p$ = .001. These analyses yield results consistent with the proposition that lyrically simpler songs enjoy greater success in time periods in which more novel song choices are available.

## Forecasting

As a final step, we generated a forecast for average lyrical compressibility for four decades after the last data point in our time series. This is in keeping with recommendations by Varnum & Grossmann [38] that papers analyzing past patterns of cultural change provide forecasts for the future. These forecasts enable a test of this theoretical model against concrete future cultural trends. Using the automated ARIMA algorithm, we also identified the best function for the novel song choices data, which we used to estimate the subsequent 40 data points. In turn, we used this estimated data in conjunction with the compressibility function to forecast the further development of lyrical compressibility. Results of this model suggest that lyrical compressibility will continue to increase over the next several decades (see Fig 1).

## Discussion

Popular music lyrics have recently been used to inform work on the cultural transmission of emotional expression [14, 66], as an index of culture-level changes in self- versus other-focus [5], and as a reflection of cultural mood in respond to economic and social threats [18, 19]. But one major trend in popular music lyrics remained underexplored and unexplained—popular music lyrics are coming increasingly simple over time. We reasoned and found support for the hypothesis that increasing lyrical simplicity is associated with increasing amounts of novel music production. That is, in times when more novel music is produced, popular songs become increasingly lyrically simple.

The relationship between mean lyrical compressibility and the amount of novel music produced each year was robust. We observed significant positive associations across three operationalizations of the amount of novel song choices and the average lyrical compressibility of popular songs. Further, the relationship between amount of novel song choices and average compressibility of popular songs remained significant when including a host of ecological, socioecological, and cultural factors linked to other types of cultural change both in univariate and multivariate analyses. By and large these other variables were not significantly associated with changes in lyrical simplicity after controlling for the potentially confounding influence of temporal autocorrelation. Of note, we also observed a significant negative association between changes in pathogen prevalence and lyrical simplicity. This observation suggests a potentially new consequence of infectious disease threat, one that should be explored in more detail in future work.

Importantly, the linkage between amount of new music produced and average compressibility of popular songs also held when accounting for temporal autocorrelation using three distinct methods. Thus, results suggest that the amount of novel music produced contributes to changes in average lyrical compressibility above and beyond other plausible causes and autoregressive trends in the data.

In exploratory analyses, we also found evidence suggesting that success, as indexed by position in the billboard charts, *among* popular songs was associated with greater lyrical compressibility. This is broadly consistent with the notion that simpler content enjoys an advantage in memorability and/or transmission. Importantly, this effect appeared to be stronger in years when the amount of novel songs produced was higher, providing conceptual confirmation of our key finding. More novel song choices appear linked to both greater average lyrical compressibility of the body of songs that succeeds (i.e., those entering the billboard chart in a given year), and, among songs entering the charts in a given year, compressibility was more strongly associated with better performance on the chart in years when more novel songs were produced.

This finding might parallel ongoing research taking information-theoretic approaches in exploring communicative efficiency in human language [67, 68]. For example, in both language and music, something akin to Zipf's law seems to be at play [2]—i.e., the frequency rank of a phenomenon is inversely proportional to its probability, such that, in the case of language, many words are quite rare, but a few words (e.g., pronouns) appear with great frequency. Moreover, these more successful (i.e., frequently-used) words tend be shorter in length (but see also Piantadosi et al., 2011 [69]). This observation dovetails with our finding regarding the success of simpler lyrics. Indeed, the increasingly success of simple lyrics may reflect increasing communicative efficacy.

A preference for simpler information in increasingly information-saturated environments might also be consistent with some propositions from cultural evolutionary theory. One tenet of cumulative cultural evolutionary theory is that human innovation, transmission, and learning increase the amount and quality of cultural information, while also increasing the learnability of this information [70, 25]. One way to increase information learnability is via simplicity [71, 72], thereby yielding increasingly efficient communication.

The present report adds to two growing bodies of empirical research—work emphasizing the examination of cultural products as a window into cultural-level psychological processes [14, 5] and work using time-series methods to test hypotheses regarding the causes of particular patterns of cultural change (for a review see Varnum & Grossmann, 2017 [38]). Here, we use big data and time series methods to show that increases in the amount of novel songs over time appear to be linked to the increasing simplicity of popular songs' lyrics, as well as greater success of songs with simpler lyrics. What does this tell us more broadly about how American culture has changed? It suggests potentially that success of aesthetic complexity at the cultural level may be something that shifts over time. Although this is not the first such demonstration of this phenomenon, to our knowledge this is the first attempt to formally evaluate why such cultural-level preferences may change.

## Alternative and complementary explanations

Although we found that our key effect was highly robust, alternative or complementary explanations for the growing success of lyrically simpler songs are still possible. For example, changes in the ways that people consume popular music could perhaps affect lyrical simplicity. Technological innovation (e.g., various portable music devices) could play a role, as could other variation in the ways that people interact with music. Relatedly, one might speculate that the success of increasingly simple lyrics might owe to technologically mediated increases in listening to music primarily in the background (e.g., on commutes, in gyms). However, one might easily argue that for generations music has been consumed in this fashion albeit with slightly different technologies—portable radios, car stereos, and portable music players have existed and been widely used for decades. It would be interesting to attempt to assess this question empirically, although we are not currently aware of high-quality time series data relating to how and why people listen to popular music. Moreover, operationalization of these indicators of technological innovations over time would be a potentially thorny problem. For instance, what does it mean to own a Walkman in 1982 as compared to a similar device in 2002? Nonetheless, it would be intriguing to assess these questions in future work.

Another possibility is that the length of songs may have changed over time affecting average lyrical complexity. Thus, perhaps song lyrics are more compressible by virtue of songs becoming shorter. However, a recent analysis of songs entering the Billboard charts over the course of its history suggests, in fact, that the average song on the charts in the late 2010's was somewhat longer than those in the 1950's and 1960's, and similar in recent years to levels observed in the 1970's [73]. Thus, this alternative explanation cannot account for the trends observed in the present analyses.

One might alternatively speculate that the rise in lyrical simplicity observed in the present data might be related to trends in the popularity of different musical genres. Indeed, although this is beyond the scope of the present work, it would be interesting to empirically assess how lyrical complexity varies across popular music genres and whether trends within these genres over time have been similar. Further, future work might assess whether the linkage between lyrical simplicity and song success observed in our exploratory analyses varies within genres of popular music or if genres that are on average simpler enjoy greater success in times of more music production.

## Limitations

It is worth noting that our analysis was restricted to a single type of cultural product. It might be the case that empirical analysis of other domains might show similar trends and a similar relationship between amount of novel content and success of simpler content, or it may be

that different dynamics are observed when considering television shows, videogames, or other types of cultural products. For example, many have argued that television shows have become more complex and intellectually stimulating in the past few decades, entering the so-called "Golden Age of Television." However, empirical work examining complexity over time in other types of cultural products, including movies, news broadcasts, print newspapers, novels, and political speech suggests that there is in fact a broad trend toward simpler content being increasingly preferred, at least when it comes to the language used in these products [74]. It is noteworthy that Jordan and colleagues (2019) used a different measure of complexity, in this case use of a specific set of words indicate cognitive complexity, and that they find that the strength of the decline in complexity varies across different types of cultural products. Hence, future research may attempt to conceptually replicate our work by assessing compressibility of other types of cultural products over time and whether the success of such products is linked to the number of options or alternatives within that domain.

It is also worth noting that, in the present work, we assessed the simplicity of lyrics. Songs might be complex or simple in other ways as well, in terms of rhythm, melody, number of instruments played, and so on. Analyses of these features is beyond the scope of the present work, but it would be interesting to see the extent to which similar or divergent patterns are observed in these facets of successful popular music over time.

Our analysis was also limited to songs that were relatively successful over time—i.e., those that made the Billboard Hot 100 chart. This sample is quite large ($N > 14,000$), but it may not be representative of all songs produced during this period. Further, we were able to successfully scrape a greater proportion of more recent rather than older songs, which we included in control analyses. Our sample captures a large chunk of popular music produced during more than half a century and enables tests regarding linkages between novel music choices, lyrical simplicity, and song success. A slightly different conceptual question may be worthwhile addressing in future work: Does average complexity of *all* music produced change along with shifts in the amount of music produced?

Our work is also limited by the fact that song success was operationalized by commercial success in the US market. Although some cultural shifts in the past several decades appear to be global in nature, such as rising individualism [36], this need not be the case for all dimensions of culture. Different dynamics may potentially be observed in terms of song success in parts of the world with different values, practices, and ecological conditions. Although such an endeavor is beyond the scope of the present manuscript largely due to the lack of equally rich time series data from other countries, it would be worthwhile to try to address this question in the future.

Finally, the present work is limited by its correlational nature. Although our findings appeared quite robust across different operationalizations of the independent variable—when accounting for autocorrelation in various ways, and when controlling for a host of plausible ecological, socioecological factors, and cultural values which have shifted over time—we cannot completely rule out all alternative explanations for increasing success of songs with simpler lyrics. Future work might attempt to quantify society level time series trends in conformity or other biases linked to lyrical affect and music sampling [14, 75], and assess whether the present findings hold when controlling for these variables as well. Future work may also use in-lab methods to explore and disentangle the possible causal mechanisms underlying the link between amount of novel song choices and success of songs with simpler lyrics. For example, transmission chain methods [76] could be employed to explore whether participants might find simpler lyrics more pleasing and memorable when there is a greater number of other song-snippets competing for attention versus when there is not.

## Conclusion

Why have the lyrics of pop songs become simpler over time? Our findings suggest that the answer may have to do with the proliferation of new songs available to consumers. The present work represents one of the first attempts to use big data and time series methods to quantify temporal shifts in information transmission dynamics at the societal level. Future work may attempt to replicate and extend these findings into other types of complexity and other types of cultural products.

## Supporting information

**S1 Fig. Zero-order Kendall's Tau correlations between variables.**
(TIF)

**S1 File.**
(DOCX)

## Author Contributions

**Conceptualization:** Michael E. W. Varnum, Jaimie Arona Krems.

**Data curation:** Colin Morris, Alexandra Wormley.

**Formal analysis:** Michael E. W. Varnum, Colin Morris, Igor Grossmann.

**Investigation:** Michael E. W. Varnum, Colin Morris, Igor Grossmann.

**Methodology:** Michael E. W. Varnum, Colin Morris, Igor Grossmann.

**Resources:** Colin Morris.

**Validation:** Alexandra Wormley.

**Visualization:** Alexandra Wormley, Igor Grossmann.

**Writing – original draft:** Michael E. W. Varnum, Jaimie Arona Krems, Colin Morris, Igor Grossmann.

**Writing – review & editing:** Michael E. W. Varnum, Jaimie Arona Krems, Colin Morris, Alexandra Wormley, Igor Grossmann.

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
