## [Decision Letter · Decision Letter 0]

17 Sep 2020

PONE-D-20-20631

People prefer simpler content when there are more choices: A time series analysis of lyrical complexity in six decades of American popular music

PLOS ONE

Dear Dr. Varnum,

Thank you for submitting your manuscript to PLOS ONE. After careful consideration, we feel that it has merit but does not fully meet PLOS ONE’s publication criteria as it currently stands. 

The two reviewers provide constructive and partially overlapping comments on your framing and the analyses. I strongly encourage you to consider the additional analyses and validity checks proposed by reviewer 1 as well as addressing the conceptual questions raised by both reviewers 1 and 2.

I am also wondering whether genre and the proliferation and diversification of genres over the last century may partially be responsible for some of these effects. To what extent do these trends occur within genres or over the careers of artists/groups? Do novel genres have an advantage over more established genres? Greater attention to genres of music as well as trends for the same agent (singer/songwriter, performer) may help to address some of the conceptual issues identified by the reviewers.

Therefore, we invite you to submit a revised version of the manuscript that addresses the points raised during the review process.

We look forward to receiving your revised manuscript.

Kind regards,

Ronald Fischer

Academic Editor

PLOS ONE

Journal Requirements:

3. Please remove your figures from within your manuscript file, leaving only the individual TIFF/EPS image files, uploaded separately.  These will be automatically included in the reviewers’ PDF.

Additional Editor Comments (if provided):

This is an innovative and thought provoking article. The two reviewers provide constructive and partially overlapping comments on your framing and the analyses. I strongly encourage you to consider the additional analyses and validity checks proposed by reviewer 1 as well as addressing the conceptual questions raised by both reviewers 1 and 2.

I am also wondering whether genre and the proliferation and diversification of genres over the last century may partially be responsible for some of these effects. To what extent do these trends occur within genres or over the careers of artists/groups? Do novel genres have an advantage over more established genres? Greater attention to genres of music as well as trends for the same agent (singer/songwriter, performer) may help to address some of the conceptual issues raised.

Reviewers' comments:

Reviewer's Responses to Questions

**Comments to the Author**

1. Is the manuscript technically sound, and do the data support the conclusions?

Reviewer #1: Partly

Reviewer #2: Yes

2. Has the statistical analysis been performed appropriately and rigorously? 

Reviewer #1: Yes

Reviewer #2: Yes

3. Have the authors made all data underlying the findings in their manuscript fully available?

Reviewer #1: Yes

Reviewer #2: Yes

4. Is the manuscript presented in an intelligible fashion and written in standard English?

Reviewer #1: Yes

Reviewer #2: Yes

5. Review Comments to the Author

Reviewer #1: This study explores a trend towards greater compressibility of US song lyrics, which became more repetitive over the last 6 decades. The authors test the claim that this trend is due to an increase in the variety of songs on offer. The results show that novelty in music production (henceforth "musical novelty") is a significant predictor of lyrics compressibility, even when controlling, separately, for temporal autocorrelation on the one hand, and for a host of potential confounds on the other hand.

This is an exciting and innovating study, correctly done overall, and demonstrating an intringuing and non-trivial phenomenon: song lyrics become more repetitive over time. The use of a future-oriented predictive model is particularly appreciated. If the paper merely demonstrated and explored this trend I would have no reservations about it. My main concern comes from the causal hypothesis that the study puts forward to explain the trend.

The results only partially support the authors' claims. First, because the study fails to test a set of competing explanations that seem more plausible to me than the one put forward. They are detailed below. Second, because the claim that novel music production predicts lyric compressibility above other predictors (p. 18, "the amount of novel music produced contributes to changes in average lyrical compressibility above and beyond other plausible causes") is not demonstrated or even suggested by the data. Third, no evidence is given for the contention that more compressible songs are more likely to be successful, when there is more choice (in the authors' own data or elsewhere).

1. Alternative explanations

An explanation that is alluded to in one paragraph of the discussion (p. 21) but not followed through is that song lyrics became simpler and more repetitive because listening to music became something that people did while doing other things and often without paying any particular attention (in supermarkets, elevators, bars, etc., no longer just concert halls or standing on street corners). This would readily explain why lyrics become simpler: because songs no longer have the listeners' undivided attention. This explanation is entirely distinct from the hypothesised effect of musical novelty: it is about changes in music consumption, not about changes in music production. Even so, it is coherent with the pattern of results presented here. Arguably the musical industry produced increasingly many songs because demand grew, and demand grew because people took to listening to music in circumstances where they did not use to. Changes in media of diffusion (e.g. from sheet music to radio) are an obvious and related explanation. Unless we assume that these two hypotheses are somehow equivalent or interchangeable, one cannot claim that growing musical novelty caused the observed trend without ruling out this alternative account.

One may also worry about a possible selection bias. As explained in the supplementary materials, the study selected roughly half the songs that appeared in the charts for textual analysis, due to difficulties in finding good textual data for other songs. This raises the possibility that a selection bias might explain the observed trend. It is possible that text data is better for later songs: that our documentation for 2000s hits is better than it is for 1960s hits. It is possible that songs with more less compressible lyrics are more likely to be documented, because they are more interesting, lyrics-wise, and more worthy of attention. If these two conditions obtained they would suffice to produce an apparent decrease in compressibility that would be entirely due to a preservation bias. Lyric compressibility would not actually decrease through time for unrecorded song lyrics. I am not saying that this is what happened, but this explanation is easy to rule out (just show that the proprotion of hit songs with undocumented lyrics does not change through time, or that such changes, if they occur, do not explain away the trend you observe). Relatedly, more detail on the selection of song lyrics to be analysed would be welcome: what the criteria for inclusion were, whether there was any stopping rule for data collection, etc.

2. Is novel music production a better predictor of lyric compressibility than other predictors?

The results do not establish that musical novelty is a better predictor of lyrics compressibility compared to other possible predictors studied here. Several indicators show a higher correlation with lyrics compressibility, among them (judging by Fig. 1) GDP per capita, population size, and (with an inverse correlation) residential mobility. (Although I don't know what would happen to these correlations after autocorrelation is taken into account.) To sustain the claim that musical novelty is a better predictor of lyric compressibility than other candidates, running partial correlations is not sufficient. Partial correlations merely show that the correlation between lyrics compressibility and musical novelty is robust when variable X is taken into account, but it could still be the case that variable X does better, as a predictor of lyrics compressibility, than musical novelty does.

Relatedly, it is not clear whether the correlation between lyrics complexity and musical novelty would still hold once all important confounds are controlled for *together*, and not just separately as done here. The choice of analysis that was made for this study (taking years as data points) does not allow this to be shown (too few data points), but a nested regression taking songs as data points instead of years might allow the authors to demonstrate this (with due attention being paid to multicollinearity). Alternatively, the authors could reduce all the potential confounds (all factors listed in Fig. 1 except Lyric compressibility, Music production, and Year) to one super-factor, with a PCA. Showing that the correlation between lyrics complexity and musical novelty holds when doing a partial correlation controlling for this super-factor would help make the authors' point.

3. Missing evidence of greater success for simpler songs

On p. 3–4, the study justifies the hypothesis to be tested on the grounds that people generally prefer simpler content to more complex content, especially when the choice is broad. This debatable claim is made by analogy with results in social psychology and experimental economics which in my view are not clearly relevant to the material being studied here. The similarity between a simple economic decision (e.g. a financial product that is easy to understand, as in Iyengar & Kamenica 2010) and a repetitive song, seems quite remote to me. Still, this view makes one clear prediction: more compressible songs should be more commercially successful than compressible ones, at least when there is a lot of choice. The paper seems to endorse this point but does not cite any evidence for it. It would be easy to answer this question, by comparing billboard hit songs with non-hits and controlling for various other factors.

Minor comments:

One possible confound that is (in my view) unlikely to explain the study's correlations but is easy to control for and should be ruled out, is song length: given the measurement of compressibility, I suspect song length will strongly impact compressibility, and if there is any trend in time towards shorter or longer song this might confound the observed trends.

The legend for figure 1 says that the correlations between variables are given as Kendall's tau, but I doubt it for two reasons. 1: The value given in the figure for the correlation between the Music Production index and Lyric Compressibility is .88, which does not correspond to the value reported in the main text (Kendall’s τ = .714), but does correspond to the Pearson's r correlation given in the markdown file (Pearson's r = .87723). 2. In the source code for the figure the method for the correlation is not specified (the command is cor(years, use="" ext-link-type="uri" xlink:type="simple">pairwise.complete.obs")). I suspect R defaults to method = "pearson" when method isn't specified. Please clarify and correct if needed.

Correlations are occasionally (exceptionally) given using Pearson's r (p. 10, also p. 14 when reporting the results for Tiokhin-Hruschka method). The authors note that this parametric correlation is inappropriate since time-series data are not normally distributed. Please remove mentions of Pearson's r or uses of it in reporting results. I recommend paying special attention to results on the Tiokhin-Hruschka method when doing so. See also the above comment regarding Fig. 1.

p. 16 AIC stands for Akaike's Information criterion (not Aikeke).

p. 20 This passage of the discussion alludes to a section of the supplementary materials that I could not find: "the aim of the present work was to understand what shapes the success of cultural products over time, rather than to use the broadest possible set of cultural products as a way to gain insight into other phenomena at the population level (see supplement for an extended discussion of this issue)."

Reviewer #2: This paper presents an analysis of why pop music in the US has become lyrically simpler over time, testing the hypothesis that the trend is driven by an expansion in the number of available song choices. This is tested by quantifying lyrical simplicity using a metric of information compressibility (LZ77 compression algorithm) over thousands of songs, and correlating this measure with estimates of the number of new songs in each year. The results support the hypothesis: large correlations between the measures.

The paper is well written and the analyses are sound and generally appropriately interpreted. The ‘multiverse’-style analysis approach is also helpful in that it provides converging different approaches. The results will be of interest to people in the psychology of music, cultural evolution, and the general public as well.

Here are a few suggestions for a revision:

(1) What songs are most popular and make it to Billboard is not unrelated to preferences, but also not that tight of a measure of people’s self directed-listening behaviours and preference for music, as is implied by the use of "preferences" throughout the paper. for instance, radio plays are influenced by advertisers, independently of people's preferences for songs. A tighter claim to make is that, as more music becomes available, simpler songs are more memorable and/or dispersible than more complicated ones. Whether and how this is related to claims in the manuscript about peoples’ music preferences changing based on Kahneman-esque heuristics being deployed due to increased cognitive load (Intro, pages 4 and 5) and/or interpreting these changes in lyrical trends as indicating changes in emotional expression (if this is what the abstract framing + discussion is implying? Eg. in “What does this tell us more broadly about how American culture has changed?”) is more up for debate, I think. This is an easy fix: just need to clarify the interpretation in the paper a bit more.

(2) The manuscript is clear that the correlational data doesn’t justify claims about causality, but it would be helpful to tighten up the areas where an interpretative claim is being made. Might the direction of causality be backwards? Songs that are simple could be easier to produce, so as artists realize they can produce simpler styles, maybe they produce more of them? There are plenty of other explanations here that would be good to discuss. For instance, maybe memorability is a big driver in what songs get a lot of radio plays, where memorability is a different aspect of music perception than preference.

(3) There may be some interesting parallels to be drawn between these results and ongoing research in how languages more generally are shaped by communicative efficiency (see for review: Gibson et al., 2019, TICS). Namely, the primary measure of simplicity of lyrics is sensitive to word length. Zipf’s law describes the frequency structure of words in a language as being related to word length (eg, Piantadosi, 2014, Psychonomic Bulletin Review), although more recent work shows that information content of words is a better predictor of word length than frequency-rank (Piantadosi et al., 2011, PNAS): in other words, more predictable words tend to be shorter. Something like Zipf's law is at work in music (see Levitin et al., 2012, PNAS; Mehr et al., 2019, Science) and so this connection with information-theoretic notions of communication would be productive. (It also fits neatly with how lyrical simplicity is quantified with LZ77).

(4) To what extent is variance in lyrical compressibility in these data mediated by the distribution of genres within the presented dataset? Electronic/dance music often has highly simple repetitive lyrics as a defining feature, for example, more so than, e.g., jazz lyrics. Perhaps one of the reasons for the popularity of electronic/dance genres within the broader popular music space may relate to this claimed attraction toward simplicity of lyrics. But the deeper point is then to ask how much of the variance in lyrical compressibility is stemming from a general trend across popular music genres and how much is contributed by relative shifts in other stylistic factors (that may be correlated with greater lyrical compressibility for additional reasons). Disentangling this is probably difficult, but I feel like it could be discussed.

Minor comments:

For the predictions about the lyrical compressibility of future popular music, some comments about the bounds in which such extrapolation is valid/meaningful would be helpful. What does it mean for music to have an average compressibility index of ~1.225 by 2050 (as compared to the current average of ~1.1)? What are reasonable bounds of compressibility that things might plateau at?

Please check references, as at least one in-text citation was not in the end references (Steegen et al., 2016)

Mehr Krasnow 2017 is a bit of a funny citation for "music is a human universal". I think better might be Mehr et al., 2019, Science and/or the new BBS theoretical treatment (https://doi.org/10.1017/S0140525X20000345)

A reference about how lyrics play an important part in people’s listening habits may be helpful. For instance, this paper based on Spotify listening data would be a helpful citation: http://archives.ismir.net/ismir2018/paper/000098.pdf.

6. PLOS authors have the option to publish the peer review history of their article (what does this mean?). If published, this will include your full peer review and any attached files.

Reviewer #1: No

Reviewer #2: No

---

## [Author Response · Author response to Decision Letter 0]

15 Oct 2020

Dear Dr. Fischer,

We appreciate your inviting the revision of manuscript, now entitled “Why are song lyrics becoming simpler? A time series analysis of lyrical complexity in six decades of American popular music.”

To remind you of the contribution, briefly, we explore the surprising trend that popular songs are becoming increasingly simple. We reason that the increasing production of novel songs may drive this phenomenon and test this association, finding a robust link. We situate this finding in the growing bodies of work using song lyrics to assess culture-level phenomena and work using time series analysis to understand drivers of cultural change. 

We see this work as being of interest to not only to those interested in social or cultural psychology, but also those studying communication, cognitive science, and music, as well as to the lay public. 

Below, we detail the changes made to this revision in line with the reviews, point-by-point, including a significant number of additional analyses. You will find critiques in plain text, with our replies italicized below. We have also highlighted major changes in the revised manuscript file in yellow for your convenience.

Reviewer 1 

This is an exciting and innovating study, correctly done overall, and demonstrating an intriguing and non-trivial phenomenon. 

We thank the reviewer for their enthusiasm for the work. 

The results only partially support the authors' claims. First, because the study fails to test a set of competing explanations that seem more plausible to me than the one put forward. They are detailed below. Second, because the claim that novel music production predicts lyric compressibility above other predictors (p. 18, "the amount of novel music produced contributes to changes in average lyrical compressibility above and beyond other plausible causes") is not demonstrated or even suggested by the data. Third, no evidence is given for the contention that more compressible songs are more likely to be successful, when there is more choice (in the authors' own data or elsewhere).

We have run a significant number of new analyses to address this. In particular, we comprehensively address the reviewer’s second and third points, finding that novel song production is a robust predictor of lyrical simplicity even over and above a host of other ecological and cultural predictors (see Tables 1 and 2), including in new multivariate analyses (see page 19 “Multivariate analyses”), and showing new evidence that this relationship between song success per se (as indexed by a song’s position on the Billboard chart) and novel song production is strongest in years when there are more novel songs produced (see pages 19-21 “Exploratory song-level analyses”). 

 The reviewer also raised two competing hypotheses: 

An explanation that is alluded to in one paragraph of the discussion (p. 21) but not followed through is that song lyrics became simpler and more repetitive because listening to music became something that people did while doing other things and often without paying any particular attention (in supermarkets, elevators, bars, etc., no longer just concert halls or standing on street corners). This would readily explain why lyrics become simpler: because songs no longer have the listeners' undivided attention. This explanation is entirely distinct from the hypothesised effect of musical novelty: it is about changes in music consumption, not about changes in music production. 

We now address this point at even greater length in the discssion section (see page 25 first full paragraph), noting that, in particular, technology-mediated changes may influence music consumption practices. However, respectfully disagree that changes in listener attention are likely to cause the shift in lyrical simplicity seen here; for example, people have listened to music in their cars for decades, portable music players have been available for decades, and music has been featured as the background noise in various entertainment establishments for decades. Further, although an interesting avenue for future research, we feel it is beyond the scope of the present work to assess music listening patterns for reasons described on page 25 first full paragraph)

One may also worry about a possible selection bias. As explained in the supplementary materials, the study selected roughly half the songs that appeared in the charts for textual analysis, due to difficulties in finding good textual data for other songs. This raises the possibility that a selection bias might explain the observed trend. It is possible that text data is better for later songs: that our documentation for 2000s hits is better than it is for 1960s hits. It is possible that songs with more less compressible lyrics are more likely to be documented, because they are more interesting, lyrics-wise, and more worthy of attention.

In order to address this point we conducted analyses that controlled for percentage of charting songs for which lyrics could be successfully scraped (see page 18-19 “Robustness analyses: Controlling for percentage of scraped songs”). Our key relationship held controlling for this possibility.

Relatedly, more detail on the selection of song lyrics to be analysed would be welcome: what the criteria for inclusion were, whether there was any stopping rule for data collection, etc.

Additional details regarding the processing of song lyrics can be found on pages 2-3 of the Supporting Information.

Is novel music production a better predictor of lyric compressibility than other predictors?...  The results do not establish that musical novelty is a better predictor of lyrics compressibility compared to other possible predictors studied here. Several indicators show a higher correlation with lyrics compressibility, among them (judging by Fig. 1) GDP per capita, population size, and (with an inverse correlation) residential mobility. (Although I don't know what would happen to these correlations after autocorrelation is taken into account.) To sustain the claim that musical novelty is a better predictor of lyric compressibility than other candidates, running partial correlations is not sufficient. Partial correlations merely show that the correlation between lyrics compressibility and musical novelty is robust when variable X is taken into account, but it could still be the case that variable X does better, as a predictor of lyrics compressibility, than musical novelty does.

We understand the reviewer’s concern, however we note that we do not claim that musical novelty is the best predictor of average lyrical compressibility. That said, we believe that new analyses in which we look at detrended relationships between all putative predictors and average lyrical compressibility suggest that it is one of only three significant predictors, and the only one for which we frankly had an a priori hypothesis when we began the work. We attempt no interpretation of the negative relationship between conservatism and compressibility, and we do talk briefly about the negative relationship between pathogens and compressibility, which we suggest should be followed up on in the future. That said, again our focus was on testing our a priori hypotheses about ONE possible driver of growing lyrical simplicity, hence we focus on this in the present manuscript.

Relatedly, it is not clear whether the correlation between lyrics complexity and musical novelty would still hold once all important confounds are controlled for *together*, and not just separately as done here. The choice of analysis that was made for this study (taking years as data points) does not allow this to be shown (too few data points), but a nested regression taking songs as data points instead of years might allow the authors to demonstrate this (with due attention being paid to multicollinearity). Alternatively, the authors could reduce all the potential confounds (all factors listed in Fig. 1 except Lyric compressibility, Music production, and Year) to one super-factor, with a PCA. Showing that the correlation between lyrics complexity and musical novelty holds when doing a partial correlation controlling for this super-factor would help make the authors' point. 

We are grateful to the reviewer for this suggestion. Our new multivariate analyses follow these suggestions (p.19) and find that our key effect holds. Taken together we believe we have a great deal of evidence for the robustness of our key finding and we are grateful to the reviewer for helping strengthen the rigor of the manuscript.

 Missing evidence of greater success for simpler songs  On p. 3–4, the study justifies the hypothesis to be tested on the grounds that people generally prefer simpler content to more complex content, especially when the choice is broad. This debatable claim is made by analogy with results in social psychology and experimental economics which in my view are not clearly relevant to the material being studied here. The similarity between a simple economic decision (e.g. a financial product that is easy to understand, as in Iyengar Kamenica 2010) and a repetitive song, seems quite remote to me. Still, this view makes one clear prediction: more compressible songs should be more commercially successful than compressible ones, at least when there is a lot of choice. The paper seems to endorse this point but does not cite any evidence for it. It would be easy to answer this question, by comparing billboard hit songs with non-hits and controlling for various other factors. 

Great point! We’ve taken this advice to heart (see pages 19-20, “Exploratory Song-level analyses,”) and we do find empirical support for this claim. Namely, among Billboard charting songs, those that are more compressible are more successful. Further this relationship is stronger in years in which more novel songs are produced. We thank the reviewer for suggesting this and we believe again that the rigor of the manuscript and the fit between evidence and the rationale in the introduction has been enhanced as a result.

Minor comments

 - One possible confound that is (in my view) unlikely to explain the study's correlations but is easy to control for and should be ruled out, is song length: given the measurement of compressibility, I suspect song length will strongly impact compressibility, and if there is any trend in time towards shorter or longer song this might confound the observed trends.

We now address this possibility in the discussion section. Based on empirical findings regarding song length of Billboard charting songs, we do not feel that this alternative explanation can explain our observations. See below (from page 26): 

“Another possibility is that the length of songs may have changed over time affecting average lyrical complexity. Thus, perhaps song lyrics are more compressible by virtue of songs becoming shorter. However, a recent analysis of songs entering the Billboard charts over the course of its history suggests, in fact, that the average song on the charts in the late 2010’s was somewhat longer than those in the 1950’s and 1960’s, and similar in recent years to levels observed in the 1970’s (Bannister, 2017). Thus, this alternative explanation cannot account for the trends observed in the present analyses.” 

- The legend for figure 1 says that the correlations between variables are given as Kendall's tau, but I doubt it for two reasons. 1: The value given in the figure for the correlation between the Music Production index and Lyric Compressibility is .88, which does not correspond to the value reported in the main text (Kendall’s τ = .714), but does correspond to the Pearson's r correlation given in the markdown file (Pearson's r = .87723). 2. In the source code for the figure the method for the correlation is not specified (the command is cor(years, use="pairwise.complete.obs")). I suspect R defaults to method = "pearson" when method isn't specified. Please clarify and correct if needed. 

We are grateful to the reviewer for catching this error. This has now been corrected in Table S1 which reports kendall’s tau’s instead of pearson’s r’s.

Correlations are occasionally (exceptionally) given using Pearson's r (p. 10, also p. 14 when reporting the results for Tiokhin-Hruschka method). The authors note that this parametric correlation is inappropriate since time-series data are not normally distributed. Please remove mentions of Pearson's r or uses of it in reporting results. I recommend paying special attention to results on the Tiokhin-Hruschka method when doing so. See also the above comment regarding Fig. 1.

 We understand the reviwer’s concern here. However we note that the Tiokhin-Hruschka procedure can only produce corrected significance thresholds for Pearon’s r at present. We have opted to leave these results in in the spirit of a multiverse approach. Importantly, this is only one approach used to account for autocorrelation, and importantly we get converging inferences using these different approaches. However, if the editor wishes, we are happy to move this section the supplement or to OSF as a supporting file.

 p. 16 AIC stands for Akaike's Information criterion (not Aikeke). 

Again, we are grateful to the reviewer for catching the error. It is now corrected. 

p. 20 This passage of the discussion alludes to a section of the supplementary materials that I could not find: "the aim of the present work was to understand what shapes the success of cultural products over time, rather than to use the broadest possible set of cultural products as a way to gain insight into other phenomena at the population level (see supplement for an extended discussion of this issue)."

We discuss this issue on page 27-8 of the revised manuscript and on pages 3-4 of the revised supplement. We hope this discussion is sufficient.

Reviewer 2 

The paper is well written and the analyses are sound and generally appropriately interpreted. The ‘multiverse’-style analysis approach is also helpful in that it provides converging different approaches. The results will be of interest to people in the psychology of music, cultural evolution, and the general public as well.

We thank the reviewer for their enthusiasm for the work.  

What songs are most popular and make it to Billboard is not unrelated to preferences, but also not that tight of a measure of people’s self directed-listening behaviours and preference for music, as is implied by the use of "preferences" throughout the paper. for instance, radio plays are influenced by advertisers, independently of people's preferences for songs. A tighter claim to make is that, as more music becomes available, simpler songs are more memorable and/or dispersible than more complicated ones. Whether and how this is related to claims in the manuscript about peoples’ music preferences changing based on Kahneman-esque heuristics being deployed due to increased cognitive load (Intro, pages 4 and 5) and/or interpreting these changes in lyrical trends as indicating changes in emotional expression (if this is what the abstract framing + discussion is implying? Eg. in “What does this tell us more broadly about how American culture has changed?”) is more up for debate, I think. This is an easy fix: just need to clarify the interpretation in the paper a bit more.

We have addressed this issue in line with the reviewer’s helpful comment; namely we clarify the interpretation in the present revision. 

The manuscript is clear that the correlational data doesn’t justify claims about causality, but it would be helpful to tighten up the areas where an interpretative claim is being made. Might the direction of causality be backwards? Songs that are simple could be easier to produce, so as artists realize they can produce simpler styles, maybe they produce more of them? There are plenty of other explanations here that would be good to discuss. For instance, maybe memorability is a big driver in what songs get a lot of radio plays, where memorability is a different aspect of music perception than preference.

We agree that causal inference is inherently limited when analyzing this type of data. We have tried throughout the revised manuscript to be cautious in terms of causal and mechanistic claims, especially in the revised discussion section. We have also added several new analyses (see replies to reviewer 1 for details) that we hope do strengthen the inferences made, although again stopping short of claiming to show causality.

There may be some interesting parallels to be drawn between these results and ongoing research in how languages more generally are shaped by communicative efficiency (see for review: Gibson et al., 2019, TICS). Namely, the primary measure of simplicity of lyrics is sensitive to word length.

 Zipf’s law describes the frequency structure of words in a language as being related to word length (eg, Piantadosi, 2014, Psychonomic Bulletin Review), although more recent work shows that information content of words is a better predictor of word length than frequency-rank (Piantadosi et al., 2011, PNAS): in other words, more predictable words tend to be shorter. Something like Zipf's law is at work in music (see Levitin et al., 2012, PNAS; Mehr et al., 2019, Science) and so this connection with information-theoretic notions of communication would be productive. (It also fits neatly with how lyrical simplicity is quantified with LZ77). 

We thank the reviewer for pointing out this interesting parallel, which we now treat at some length in the Discussion (pages 23-24). We additionally link the present data and this work to another area of literature dealing with cultural evolution and communicative efficiency: 

Minor comments: For the predictions about the lyrical compressibility of future popular music, some comments about the bounds in which such extrapolation is valid/meaningful would be helpful. What does it mean for music to have an average compressibility index of ~1.225 by 2050 (as compared to the current average of ~1.1)? What are reasonable bounds of compressibility that things might plateau at?

We are grateful for this insightful set of suggestions. We have now added the following description which we hope helps guide the reader’s intuitions (pg. 7) : “A score of 0 means no compression was possible (e.g. if the input were random noise), a score of 1 means a 50% reduction in size, a score of 2 means a 75% reduction, and so on.”

Further, there is a theoretical upper limit on compressibility score for any given length. The most repetitive possible song of length n would be a single letter repeated n times, and it would have a score of (log n) - 2. But this is so far from the reality of the data as to not be very interesting.

Please check references, as a least one in-text citation was not in the end references (Steegen et al., 2016). 

We have now double checked the reference list and it should now match all in text citations. Thanks to the reviewer for catching this!

Mehr Krasnow 2017 is a bit of a funny citation for "music is a human universal". I think better might be Mehr et al., 2019, Science and/or the new BBS theoretical treatment (https://doi.org/10.1017/S0140525X20000345)

We agree and have switched the citation to Mehr et al., 2019. 

AE Decision Letter

I am also wondering whether genre and the proliferation and diversification of genres over the last century may partially be responsible for some of these effects. To what extent do these trends occur within genres or over the careers of artists/groups? Do novel genres have an advantage over more established genres? Greater attention to genres of music as well as trends for the same agent (singer/songwriter, performer) may help to address some of the conceptual issues identified by the reviewers.

These are good points. We agree that genre would be an interesting avenue for future exploration and we now include an extended discussion of this issue in the revised discussion section (page 26). In terms of tracking the course of an individual artist’s output, this would also be an intriguing possibility, however we would be dealing with small N’s for most and potential confounds having to do with the aging process (i.e. executive function decline with age) that would be difficult to disentangle from broader cultural forces. We hope that with the additional analyses, revisions, and explication now provided that the reviewers points are largely addressed even though we did not opt to attempt analyses by genre or within artist. We hope that you will agree that the new analyses reported in the revision are in fact sufficient to all most major concerns.

In sum, we believe that we have addressed all major points raised by reviewers, and that the present revision is suitable for publication in PLOS ONE. We are grateful to the two reviewers and to yourself for the insightful feedback and critique. We believe the manuscript has improved tremendously as a result. We look forward to your reply.

Sincerely,

Michael E. W. Varnum

---

## [Decision Letter · Decision Letter 1]

14 Dec 2020

Why are song lyrics becoming simpler? A time series analysis of lyrical complexity in six decades of American popular music

PONE-D-20-20631R1

Dear Dr. Varnum,

We’re pleased to inform you that your manuscript has been judged scientifically suitable for publication and will be formally accepted for publication once it meets all outstanding technical requirements.

Kind regards,

Ronald Fischer

Academic Editor

PLOS ONE

Additional Editor Comments (optional):

Congratulations, I recommend your article for publication to the Editor in Chief.

Reviewers' comments:

Reviewer's Responses to Questions

**Comments to the Author**

1. If the authors have adequately addressed your comments raised in a previous round of review and you feel that this manuscript is now acceptable for publication, you may indicate that here to bypass the “Comments to the Author” section, enter your conflict of interest statement in the “Confidential to Editor” section, and submit your "Accept" recommendation.

Reviewer #1: All comments have been addressed

Reviewer #2: All comments have been addressed

2. Is the manuscript technically sound, and do the data support the conclusions?

Reviewer #1: Yes

Reviewer #2: Yes

3. Has the statistical analysis been performed appropriately and rigorously? 

Reviewer #1: Yes

Reviewer #2: Yes

4. Have the authors made all data underlying the findings in their manuscript fully available?

Reviewer #1: Yes

Reviewer #2: Yes

5. Is the manuscript presented in an intelligible fashion and written in standard English?

Reviewer #1: Yes

Reviewer #2: Yes

6. Review Comments to the Author

Reviewer #1: All my comments were addressed more than satisfactorily. The authors are to be congratulated for this excellent contribution!

Reviewer #2: (No Response)

7. PLOS authors have the option to publish the peer review history of their article (what does this mean?). If published, this will include your full peer review and any attached files.

Reviewer #1: **Yes: **Olivier Morin

Reviewer #2: No

---

## [Editor Report · Acceptance letter]

18 Dec 2020

PONE-D-20-20631R1 

Why are song lyrics becoming simpler? A time series analysis of lyrical complexity in six decades of American popular music 

Dear Dr. Varnum:

I'm pleased to inform you that your manuscript has been deemed suitable for publication in PLOS ONE. Congratulations! Your manuscript is now with our production department. 

Kind regards, 

on behalf of

Dr. Ronald Fischer 

Academic Editor

PLOS ONE